# Progressive External Ophthalmoplegia in Polish Patients—From Clinical Evaluation to Genetic Confirmation

**DOI:** 10.3390/genes12010054

**Published:** 2020-12-31

**Authors:** Biruta Kierdaszuk, Magdalena Kaliszewska, Joanna Rusecka, Joanna Kosińska, Ewa Bartnik, Katarzyna Tońska, Anna M. Kamińska, Anna Kostera-Pruszczyk

**Affiliations:** 1Department of Neurology, Medical University of Warsaw, Banacha 1a, 02-097 Warsaw, Poland; amkaminska@wum.edu.pl (A.M.K.); anna.kostera-pruszczyk@wum.edu.pl (A.K.-P.); 2Institute of Genetics and Biotechnology, Faculty of Biology, University of Warsaw, Pawinskiego 5a, 02-106 Warsaw, Poland; mkaliszewska@gmail.com (M.K.); ruseckaj@gmail.com (J.R.); ebartnik@igib.uw.edu.pl (E.B.); kaska@igib.uw.edu.pl (K.T.); 3Department of Medical Genetics, Medical University of Warsaw, Pawinskiego 3c, 02-106 Warsaw, Poland; joanna.kosinska@wum.edu.pl

**Keywords:** mitochondrial disorders, progressive external ophthalmoplegia, *POLG* gene, *TWNK* gene, *RNASEH1* gene, mitochondrial DNA deletions, multiple mitochondrial DNA deletions, muscle biopsy

## Abstract

Mitochondrial encephalomyopathies comprise a group of heterogeneous disorders resulting from impaired oxidative phosphorylation (OxPhos). Among a variety of symptoms progressive external ophthalmoplegia (PEO) seems to be the most common. The aim of this study is to present clinical and genetic characteristics of Polish patients with PEO. Clinical, electrophysiological, neuroradiological, and morphological data of 84 patients were analyzed. Genetic studies of mitochondrial DNA (mtDNA) were performed in all patients. Among nuclear DNA (nDNA) genes *POLG* was sequenced in 41 patients, *TWNK* (*C10orf2*) in 13 patients, and *RNASEH1* in 2 patients. Total of 27 patients were included in the chronic progressive external ophthalmoplegia (CPEO) group, 24 in the CPEO+ group. Twenty-six patients had mitochondrial encephalomyopathy (ME), six patients Kearns–Sayre syndrome (KSS), and one patient sensory ataxic neuropathy, dysarthria, ophthalmoparesis (SANDO) syndrome. Genetic analysis of nDNA genes revealed the presence of pathogenic or possibly pathogenic variants in the *POLG* gene in nine patients, the *TWNK* gene in five patients and the *RNASEH1* gene in two patients. Detailed patients’ history and careful assessment of family history are essential in the diagnostic work-up. Genetic studies of both mtDNA and nDNA are necessary for the final diagnosis of progressive external ophthalmoplegia and for genetic counseling.

## 1. Introduction

Mitochondrial disorders result from mutations either in mitochondrial DNA (mtDNA) or nuclear DNA (nDNA). Each human cell with the exception of mature erythrocytes contains 500 to 6000 of mitochondria and each mitochondrion contains one to fifteen copies of mitochondrial DNA [1,2]. The number of mtDNA copies differs significantly among cell types and tissues and depends on the energy demands. During reproduction only maternal mitochondria are inherited. Mitochondrial DNA is a double-stranded 16-kilobase molecule encoding 13 proteins essential for oxidative phosphorylation, 22 types of tRNA, and 2 types of rRNA. All subunits of complex II as well as all other OxPhos complex proteins are encoded by nuclear DNA.

Nuclear genes are responsible for mtDNA maintenance as all proteins engaged in mtDNA replication and repair as well as for providing nucleotides for these processes are encoded by chromosomal genes. Less obviously, mtDNA stability is assured by the proper mitochondrial dynamics and thus proteins involved in fusion and fission of these organelles.

Mitochondrial genome stability may be disturbed by mutations in nuclear genes encoding mostly proteins that are part of the mitoproteome. Mitochondrial DNA replication depends on proteins such as POLG (*POLG*) and POLG2 (*POLG2*) subunits of polymerase γ—the only replicative mitochondrial DNA polymerase, mitochondrial helicase TWINKLE (*TWNK* also known as *C10orf2* or *PEO1*), mitochondrial transcription factor A (*TFAM*), or RNase H1 (*RNASEH1*) [3,4]. Besides this, for example conserved helicase/nuclease DNA2 (*DNA2*) and mitochondrial genome maintenance exonuclease (*MGME1*) influence mtDNA repair processes. dNTP pool maintenance is supported, among others, by mitochondrial ADP/ATP translocase ANT1 (*SLC25A4*), thymidine phosphorylase (*TYMP*), thymidine kinase (*TK2*), deoxyguanosine kinase (*DGUOK*), ribonucleotide reductase (e.g., *RRM2B*), succinyl-CoA synthetase (e.g., *SUCLA2*), and 4-aminobutyrate aminotransferase (*ABAT*). In addition to this, mitochondrial DNA dynamics is regulated by OPA1 (*OPA1*) and MFN2 (*MFN2*), mitochondrial inner membrane protein MPV17 (*MPV17*) and F-box and leucine rich repeat protein 4 (*FBXL4*) [4]. All these processes are involved in the pathophysiology of mitochondrial disorders because impaired mtDNA maintenance leads to the dysfunction of respiratory chain complexes and energy production [3,5]. 

Clinical and genetic variability of mitochondrial disorders provoked numerous attempts to elaborate guidelines for physicians and geneticists. Bernier et al. provided major and minor criteria for patients with respiratory chain disorders using clinical, morphological, enzymatic, and molecular results [6]. In 2006 Morava et al. described criteria for mitochondrial disorders in children and emphasized the role of additional biochemical and imaging studies [7]. Recently Mancuso et al. proposed redefinition of classical phenotypes associated with single mtDNA deletions, however, longitudinal studies are still necessary to better characterize the phenotype-genotype correlations [8]. None of the clinical symptoms is pathognomonic for mitochondrial disorders, however, some of them seem to be more specific. They include in the first place ophthalmoplegia with ptosis and additionally epilepsy, cardiomyopathy, and arrhythmias with other conduction defects. Haas et al. proposed red flags divided among neurologic, cardiovascular, ophthalmologic, gastroenterological, and all other symptoms which should indicate further diagnostic work-up [9]. Among them cerebral stroke-like lesions in nonvascular pattern, basal ganglia disease, hypertrophic cardiomyopathy with rhythm disturbances, unexplained or valproate-induced liver failure, metabolic disturbances like failure to thrive, lactic acidosis and neuromuscular symptoms are worth mentioning [9]. However, there are many other nonspecific findings that should also be taken into account. Detailed family history may provide meaningful information for physicians. Genetic diagnosis seems to be even more complicated, as OxPhos proteins are encoded by both mitochondrial and nuclear DNA. 

Here we report about Polish patients with progressive external ophthalmoplegia. Clinical, morphological, and genetic data were widely analyzed and they led to several conclusions important for both physicians and geneticists. 

## 2. Materials and Methods

### 2.1. Patients

Clinical data of 84 patients diagnosed between 1999 and 2019 in the Department of Neurology, Medical University of Warsaw were analyzed. There were 30 men (aged 11 to 78 years) and 54 women (aged 12 to 76 years). Clinical, electrophysiological, and neuroradiological data were presented. Patients were divided according to that accepted in the literature criteria of Kearns–Sayre syndrome (KSS)—PEO accompanied by retinitis pigmentosa and beginning before 20 years of age and sensory ataxic neuropathy, dysarthria, ophthalmoparesis (SANDO) syndrome. Patients with progressive external ophthalmoplegia accompanied or not by ptosis were classified as chronic progressive external ophthalmoplegia (CPEO). Those with PEO, ptosis and limb muscle weakness were diagnosed as chronic progressive external ophthalmoplegia plus (CPEO+) [10]. Cases with central nervous system involvement (like epilepsy, mental retardation, sensorineural hearing loss) were diagnosed as mitochondrial encephalomyopathy (ME).

### 2.2. Skeletal Muscle Biopsy

The open muscle biopsies (*n* = 74) were performed under local anesthesia. The samples were obtained from the biceps brachii muscle in 68 patients, from the quadriceps muscle in 5 patients and from the deltoid muscle in 1 patient. The muscle specimens were prepared for further analyses by light microscopy using the routine methods [11]. Biopsied tissue was frozen in isopentane cooled in liquid nitrogen, cut on a cryomicrotome with a slice thickness of 8 µm, and stained for a battery of routine histological and histochemical methods. Additionally, data from ultrastructural examination was available in 9 patients from the studied group. Samples for electron microscopy (EM) were fixed in glutaraldehyde, postfixed in osmium tetroxide before embedding in Spurr-embedding medium. Ultrathin sections of the selected areas were stained with uranyl acetate and counterstained with lead citrate. The samples were viewed with JEM 1200 EX2 electron microscope. 

### 2.3. Genetic Studies

Tests for most frequent mtDNA pathogenic variants m.3243A>G, m.8344A>G, m.8993T>C, and m.8993T>G were performed as described in Kierdaszuk et al. [11].

Mitochondrial DNA deletions were analyzed by three different PCRs. Presence of the so-called common deletion was analyzed as described previously by Soong and Arnheim [12]. Other deletions located in the major mtDNA arc were detected by long PCR as described previously by Gaweł et al. [13]. When possible, whole mtDNA was amplified and checked for rearrangements. Long 16 kb PCR was performed using primers mt16426F and 16425R (Zhang et al.) and the Expand Long Template PCR System (Roche, Basel, Switzerland) (reaction conditions in Appendix A) [14]. Based on the number of additional bands visible in the particular lane of an agarose gel results are assigned to single (one), double (two), and multiple (three and more) deletions (Figure 1).

For patients with confirmed multiple mitochondrial DNA deletions or strong clinical indications, analysis of coding sequence and exon-intron boundaries of chosen nuclear gene was performed. *POLG* sequence analysis was conducted as described previously by Kaliszewska et al. [15]. *POLG2, TWNK, SLC25A4,* and *TK2* coding sequence and fragments of non-coding sequence important for mRNA maturation were amplified. All PCR products were purified using GeneJET PCR Purification Kit (Thermo Fisher Scientific, Waltham, MA, USA) according to the manufacturer’s instructions. Sequencing was performed in the Laboratory of DNA Sequencing and Oligonucleotide Synthesis, Institute of Biochemistry and Biophysics, Polish Academy of Sciences. Obtained chromatograms were aligned with appropriate reference sequence: *POLG* NG_008218.1, *POLG2* NG_013029.1, *TWNK* NG 012624.1, *SLC25A4* NG_013001.1, and *TK2* NG_016862.1.

For chosen patients with mtDNA deletions and negative results of nuclear gene analysis with Sanger sequencing, whole exome sequencing was performed as described in Ploski et al. [16]. 

The study was conducted in accordance with the Declaration of Helsinki of 1975, revised in 2013. All patients provided written consent to the procedures (DNA sample taking, WES). The study spanned several years, thus four permits were obtained from the Bioethics Commission of the Medical University of Warsaw, the relevant body dealing with permits for biomedical research at the Medical University of Warsaw. These permits have the following designations: KB/156/2008, 21 OCT 2008; KB/17/R/2011, 18 MAY 2011; KB/61/A/2013, 15 MAY 2013; KB/243/2015, 1 DEC 2015.

## 3. Results

### 3.1. Phenotype Classification

The patients were divided into five groups on the basis of the following clinical criteria—ptosis, external ophthalmoplegia, limb or trunk muscle weakness, and central nervous system involvement. Additionally, criteria of Kearns–Sayre syndrome (KSS) and sensory ataxic neuropathy, dysarthria, ophthalmoparesis (SANDO) syndrome accepted in the literature were used. Chronic progressive external ophthalmoplegia (CPEO) associated with ptosis was identified in 27 patients. There were 10 men (aged 31 to 78 years at the time of diagnosis) and 17 women (aged 12 to 71 years at the time of diagnosis). The mean age of onset in this group was 31 and the mean age of diagnosis was 42 years. Chronic progressive external ophthalmoplegia plus (CPEO+) was identified in another 24 patients. There were 8 men (aged 33 to 61 years at the time of diagnosis) and 16 women (aged 30 to 76 years at the time of diagnosis). The mean age of onset in this group was 35 and the mean age of diagnosis was 50 years. The third group was composed of 26 patients with mitochondrial encephalomyopathy (ME). Apart from muscle weakness they presented signs of mental retardation (8 patients), epilepsy (1 patient), sensorineural hearing loss (9 patients), or other signs of central nervous system involvement. There were 10 men (aged 11 to 61 years at the time of diagnosis) and 16 women (aged 18 to 66 years at the time of diagnosis). The mean age of onset in this group was 24 and the mean age of diagnosis was 43 years. Six patients were diagnosed with KSS syndrome. There were 2 men (aged 14 years at the time of diagnosis) and 4 women (aged 15 to 36 years at the time of diagnosis). The mean age of onset and diagnosis was 11 and 21 years, respectively. There was one woman aged 51 years at the time of diagnosis with SANDO syndrome symptoms from 31 years of age. Positive family history was present in 4 patients with CPEO (patients III:3 and III:15, Figure 2), in 2 patients with CPEO+ (patients III:8 and III:14, Figure 2), in 4 patients with ME and in one patient with SANDO syndrome. No families from the studied group were related.

Biochemical evaluation consisted of numerous studies and one of them was the assessment of creatine kinase (CK) activity. In the majority of cases CK was within normal ranges (32–192 U/l). However, in single patients with ME it was elevated up to five times the upper normal range, in CPEO group up to 1.5 times and in CPEO+ group up to 3.5 times. Electrophysiological studies were performed in the majority but not in all patients. Nerve conduction studies found axonal and demyelinating neuropathy in one patient with CPEO+ and in one with ME. Demyelinating polyneuropathy was observed in one patient with CPEO+. Axonal sensory and motor neuropathy was observed in 2 patients with CPEO, in 2 patients with CPEO+, and in one with ME and one with SANDO. Axonal sensory neuropathy was detected in 2 patients with CPEO and in 4 with ME. Axonal motor neuropathy was diagnosed in one patient with CPEO and one with ME. Finally, demyelinating motor and sensory neuropathy was present in one patient with CPEO+. Myopathic pattern on electromyography studies (EMG) was detected in 8 patients with PEO, 13 with CPEO+, in 9 with ME, and in one with KSS. Neurogenic changes on EMG were present in 3 patients with CPEO, in 3 patients with CPEO+, and in 6 with ME. Mixed myopathic and neurogenic abnormalities were detected in one patient with CPEO and in two patients with ME. All other performed electrophysiological study results were normal. Endocrine dysfunction was connected mainly with thyroid gland abnormalities: Hypothyroidism was present in one patient with CPEO, in 7 patients with CPEO+, and in 4 patients with ME; hyperthyroidism was detected in one patient with CPEO and in one with ME. Hypoparathyroidism was recognized in one patient with ME and in one with KSS. Cardiological assessment included echocardiography, 12-lead electrocardiography and in some cases 24-h electrocardiography Holter monitoring. In the presented group of patients it revealed conduction abnormalities in 2 patients with KSS, in 3 with CPEO, and in one with CPEO+. Neuroradiological studies revealed minor and nonspecific abnormalities in magnetic resonance imaging (MRI) in 6 patients with CPEO and in 3 with CPEO+. In 10 patients with mitochondrial encephalomyopathy widespread hyperintense lesions were observed in both hemispheres on T2-weighted and FLAIR images on MRI. Additionally, in 7 patients with ME cortical and subcortical brain and cerebellum atrophy was detected on MRI. In 5 cases magnetic resonance spectroscopy (MRS) revealed elevated lactate levels, and this was mostly correlated with pathological lesions on MRI (Figure 3). In the patient with SANDO the subcortical brain atrophy was marked. Characteristics of patients with progressive external ophthalmoplegia are presented in Table 1.

### 3.2. Skeletal Muscle Biopsy

The most often biopsied muscle was biceps brachii muscle. On light microscopy ragged-red fibers (RRF) were detected in 60 out of 74 samples (Figure 4). They are considered characteristic for mitochondrial disorders, however, not pathognomonic. Additionally, cytochrome c oxidase (COX) activity was investigated in 23 cases, and fibers devoid of COX activity were found in 17 of them and were mainly correlated with RRF (Table 1). Electron microscopy assessment revealed abnormal mitochondria with irregular cristae and paracrystalline inclusions in 7 of the 9 examined samples, in one case inclusions were present in a biopsy without RRF (Figure 5).

### 3.3. Genetic Results

Basic genetic analysis including screening for m.3243A>G, m.8344A>G, m.8993T>C, and m.8993T>G mtDNA mutations was performed for 80 probands and one family member. The m. 3243A>G variant was found in two patients with mitochondrial encephalomyopathy (Table 2).

Analysis for mtDNA deletions was performed for 79 probands and 3 family members. Preferably, DNA isolated from muscle biopsy was used (71 samples) then blood (77 samples) and hair or urinary epithelium when no other tissue was available or the blood test gave unclear results (see Table 3). In the case of 79 patients (including one family member) the test performed on DNA isolated from at least one type of tissue gave positive results (Table 2 and Table 3). Similar numbers of patients had single (34) and multiple deletions (36). The proportion of single to double and to multiple deletions was similar in groups of patients with CPEO, CPEO+, and ME. Only in the smallest KSS group no multiple deletions were found.

The deletions were found in muscle biopsy over three times more frequently than in blood.

As a general rule in the case of multiple mtDNA deletions and when clinical evidence strongly supported a mitochondrial DNA maintenance defect, the *POLG* gene coding sequence was analyzed (41 subjects) enabling a final molecular diagnosis of *POLG*-related recessive syndrome in the case of 7 patients. In two additional patients possibly pathogenic, heterozygous variants were found.

In the case of 10 probands suspected of dominant inheritance, the *TWNK* coding region was sequenced and in two cases heterozygous pathogenic variants were found confirming the molecular diagnosis of an autosomal dominant mitochondrial disease. Later pathogenic *TWNK* variants were found in three family members of one of these probands, suffering from similar symptoms.

Gene by gene Sanger sequencing performed additionally for 15 subjects (Appendix A) gave negative results while whole exome sequencing performed for 16 probands revealed a possibly pathogenic homozygous *RNASEH1* variant in one patient. Later the same homozygous *RNASEH1* variant was confirmed in the sibling of the proband with similar symptoms [17].

The summary of genetic results in patients with progressive external ophthalmoplegia is presented in Table 2. The full presentation of genetic analysis is included in Appendix A.

## 4. Discussion

Different age of onset, varied clinical presentations and individual course of the disease make the diagnosis of mitochondrial disorders challenging. Progressive external ophthalmoplegia is one of the most characteristic symptoms of mitochondrial disorders. It is a slowly progressive extraocular muscle function impairment leading to severe limitation of movements. It may be the only symptom or with a variety of other features comprise the phenotype. In some cases, it may be associated with ptosis.

The described data for the population of Polish patients are similar to the findings from other studies [18] with respect to the clinical syndromes and genetic background and we did not observe any ethnic differences [19,20]. However, this is a wide analysis of Polish patients with progressive external ophthalmoplegia which provides numerous significant findings for both clinicians and geneticists. It confirms that progressive external ophthalmoplegia may occur at different ages and may be accompanied by a variety of symptoms and signs in mitochondrial disorders. Nevertheless, ophthalmoplegia is the most prominent clinical feature in mitochondrial disorders. Lee et al. reported data about 16 patients among which they distinguish a group with PEO with purely myopathic involvement [21]. From the genetic point of view it was mainly connected with single large-scale mtDNA deletion or multiple mtDNA deletions secondary to a nuclear gene defect [20,22]. In the majority of cases single mtDNA deletions seem to be sporadic [23]. Many attempts were made to predict the disease course and progression regarding the size and the location of mitochondrial DNA deletions [24]. The influence of muscle mitochondrial DNA heteroplasmy level, mitochondrial DNA deletion size, and the location of the mitochondrial DNA deletion on the phenotype was proved by Grady et al. [24]. Clinically, over the years ophthalmoplegia and ptosis are usually accompanied by weakness of other muscles. Many patients may suffer from dysphagia as well as from sensorineural hearing loss [23]. Additionally, some patients present with multi-organ involvement which includes cardiological, endocrinological, gastroenterological, and central nervous system symptoms [25]. Cardiological assessment in patients with mitochondrial disorders seems to be especially important. According to the literature, patients with CPEO may present various abnormalities mainly connected with conduction defects. However, systo-diastolic dysfunction and remodeling of the myocardium may be present and may require special treatment and monitoring [26].

The diagnostic clinical and genetic approach to mitochondrial disorders was widely discussed in the literature [27]. Because of the variety of symptoms and signs it consists of multiple investigations and procedures. Based on our data we can say that muscle biopsy and genetic assessment of mitochondrial DNA from muscle tissue are the most meaningful. Management of patients with progressive external ophthalmoplegia led us to propose a diagnostic genetic algorithm (Figure 6). Step by step evaluation of the patient provides information about diagnostic tests and underlines the role of muscle biopsy in both morphological and genetic confirmation of a mitochondrial disorder.

Detailed family history helps to determine the most likely type of inheritance and can direct further genetic tests. Second, careful physical examination of the patient indicates if symptoms meet the criteria of syndromes such as mitochondrial encephalomyopathy, lactic acidosis, and stroke-like episodes (MELAS) or Kearns–Sayre. It seems important to pay attention to the so-called red flags. Apart from progressive external ophthalmoplegia they include retinitis pigmentosa, stroke-like episodes in young people, especially those preceded by epileptic seizures, severe headaches with vomiting, and the occurrence of diabetes and hearing loss in one family. The involvement of several systems or organs, in particular skeletal muscles and the central nervous system is highly specific for mitochondrial disorders. After clinical evaluation the next step should conduct biochemical tests—creatine kinase activity and lactate level in serum or cerebrospinal fluid. Appropriate laboratory tests should also be performed to assess kidney, liver, thyroid function, as well as carbohydrate and lipid metabolism. In selected cases, the diagnosis is supported by cerebrospinal fluid examination. Further diagnostics depends on the patient’s symptoms and signs. Cardiological assessment (ECG, Holter ECG and echocardiography), ophthalmological examination, and checking for hearing loss are advised. Electrophysiological studies play an important role and they include nerve conduction studies, electromyography, and electroencephalography. Neuroimaging studies allow the assessment of central nervous system structures and magnetic resonance imaging may be supported by MRI spectroscopy. After physical examination and additional studies, the genetic tests should be performed.

Patients with a positive family history, suspected of MELAS syndrome should be analyzed for the most frequently associated point mutations. On the other hand, patients with progressive external ophthalmoplegia or Kearns–Sayre syndrome should be assessed for mtDNA rearrangements, but these changes are less commonly found in peripheral blood. In case of maternal inheritance also mtDNA sequencing should be performed. Negative genetic tests of mtDNA obtained from peripheral blood leukocytes, non-specific symptoms or diagnostic doubts are an indication for an open skeletal muscle biopsy. The muscle tissue not only enables histopathological and ultrastructural assessment, but also provides material for mitochondrial DNA analysis [28]. In our study the majority of mutations were confirmed in the muscle samples. In some cases, other tissues like urinary sediment cells, oral mucosal epithelium, or hair follicle cells may also be used [18]. After excluding the most common point mutations and mtDNA rearrangements it is possible to sequence the entire mitochondrial genome. Multiple mtDNA deletions are always an indication for nuclear gene analysis. Especially in cases of autosomal progressive external ophthalmoplegia the *POLG*, *TWNK,* and *SLC25A4* gene should be evaluated depending on recessive or dominant inheritance. Further studies, including whole exome sequencing (WES) are recommended according to the clinical and laboratory experience and availability.

WES enabled obtaining final molecular diagnosis in only one family (out of sixteen probands tested). Low efficiency of the technique could result from the characteristics of the study group—mostly adult patients in whom genetic predispositions and environmental factors acting throughout life contribute to the final presentation of the disease. WES is especially advised in cases of very rare mitochondrial disorders [29]. WES studies in large cohorts of patients provided molecular diagnosis in up to 39% of cases, which is significant comparing to other techniques [30].

However, even sophisticated DNA studies may fail to find a molecular background despite high clinical suspicion of mitochondrial disorders. It is estimated that rare mtDNA mutations may account for about 7.4% of patients with mitochondrial disorders [31]. Because of the clinical spectrum and various stages and courses of mitochondrial disorders detecting polymerase γ gene mutations in the adult population seems to be challenging. There are numerous reasons why the *POLG* gene should be considered the most important target for genetic analysis in case of mitochondrial disease and its genetic counselling [32,33]. First, more than 150 variants are regarded as pathogenic and the majority of them have been found in compound heterozygotes. Although the detailed molecular disease-causing mechanism of many of them remains mysterious [14] the general impact on mtDNA quantity and quality is known. First, the reduction of polymerase γ activity and processivity may directly lead to a decrease in mtDNA copy number called mtDNA depletion visible especially in POLG-related diseases in small children. Second, disturbed replication may result in mitochondrial genome rearrangements, especially multiple deletions. Third, pathogenic variants, in particular in regions encoding the POLG exonuclease domain, may result in introducing multiple point mutations into the newly synthesized mtDNA molecule.

Mutations in the *POLG* gene are associated with various clinical syndromes: autosomal recessive Alpers–Huttenlocher syndrome in small children, ataxia-neuropathy spectrum in teenagers and young adults, and adult PEO-related disorder which beside ocular muscles may affect skeletal muscles as well as the brain, heart, and peripheral nerves [25]. The latter, in most of the cases, shows autosomal recessive inheritance but autosomal dominant cases are also described [23]. Findings in our patients with *POLG* mutations are in agreement with data in the literature.

Another nuclear protein responsible for autosomal dominant PEO cases is TWINKLE (*TWNK*, *C10orf2*), a DNA helicase, whose gene is located at chromosome 10 [15]. Both missense mutations and an in-frame duplication of 13 amino acids were related to multiple mtDNA deletions and late-onset mitochondrial diseases [25]. The third form of autosomal dominant PEO is related to mutations in the chromosome 4 gene for adenine nucleotide translocator 1 protein (*SLC25A4*) [25].

Several phenotypes related to variants in the *TWNK* gene have been described in the literature. Infantile-onset spinocerebellar ataxia (IOSCA) is a progressive neurodegenerative disorder found in several patients mainly in Finland [34]. Cases with middle-age-onset cerebellar ataxia were also reported which broaden the spectrum of pathogenic *TWNK* mutations [35]. Another *TWNK*-related disorder is a hepatocerebral mtDNA maintenance defect, severe multisystem syndrome. Up to date over 40 pathogenic point mutations have been described in the *TWNK* gene [36]. However, it is still not known how mutations correlate with the severity of phenotype. Moreover, mutations in the *TWNK* gene were described in Perrault syndrome, a very rare autosomal recessive disease presenting with sensorineural hearing loss, ataxia, and ovarian dysfunction [37,38].

According to the literature data, the majority of mutations in the *TWNK* gene are connected with adult-onset syndromes with predominant ocular muscle manifestations. Central nervous system involvement is much less frequent. Additionally, COX-negative fibers may be present in the muscle tissue. Genetic analysis may reveal mtDNA deletions [39].

Our clinical findings in patients with *TWNK* gene mutation are correlated with literature data. Ophthalmoplegia and ptosis accompanied by dysphagia were present as described in Fratter et al. [40].

Patients with *RNASEH1* mutations were diagnosed as mitochondrial encephalomyopathy with PEO and marked cerebellar syndrome. Similar descriptions are presented in the literature, where progressive cerebellar ataxia was accompanied by dysphagia in almost 50% of the cases [19].

## 5. Conclusions

Patients with progressive external ophthalmoplegia require a wide range of investigations as well as advanced genetic testing to determine the final diagnosis. Among the nuclear genes the *POLG* gene seems to be the most commonly implicated, however, further studies are necessary in order to better characterize other nuclear genes involved in mitochondrial DNA instability.

## Figures and Tables

**Figure 1 genes-12-00054-f001:**
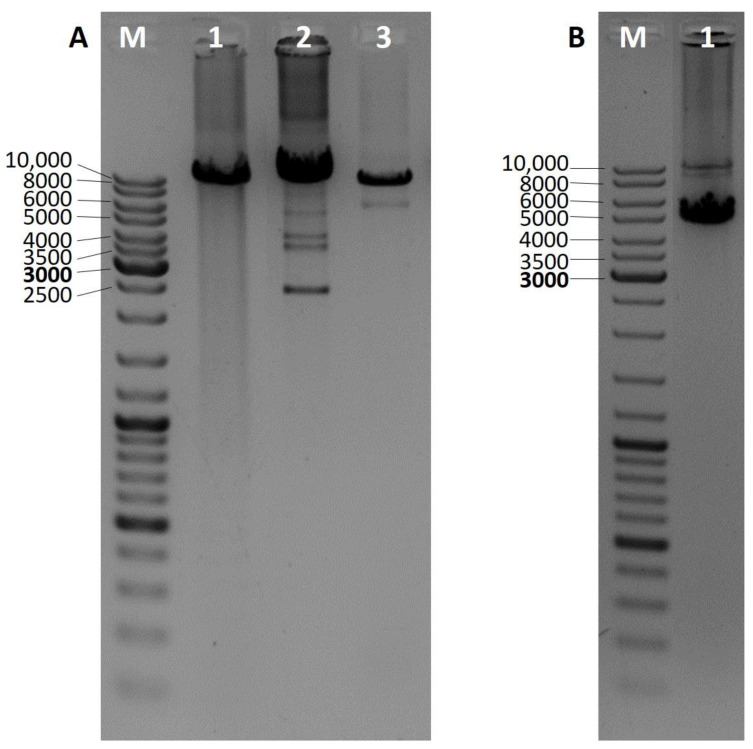
Long PCR 1% agarose gel electrophoresis results for the identification of mitochondrial DNA deletions in the so-called mtDNA major arc. Wild-type products are approximately 10 kb long, additional, shorter products indicate the presence of a mtDNA deletion. (**A**) M—DNA ladder, 1—no deletions, 2—multiple deletions, 3—single deletion; (**B**) M—DNA ladder, 1—two deletions.

**Figure 2 genes-12-00054-f002:**
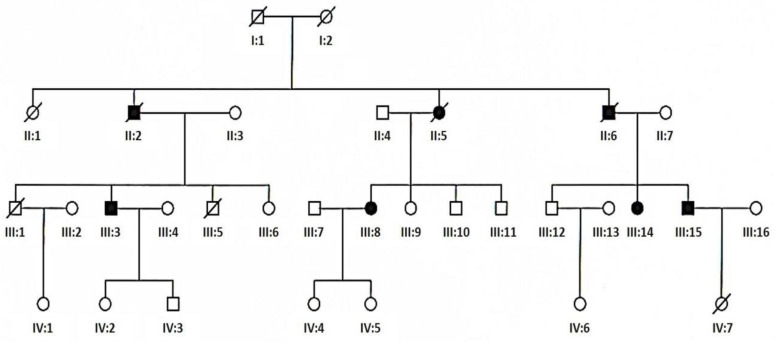
Pedigree of family with TWNK mutation (empty circle—female, empty square—male, solid circle—affected female, solid square—affected male, slashed circle—deceased female, slashed square—deceased male, slashed solid circle—deceased affected female, slashed solid square—deceased affected male).

**Figure 3 genes-12-00054-f003:**
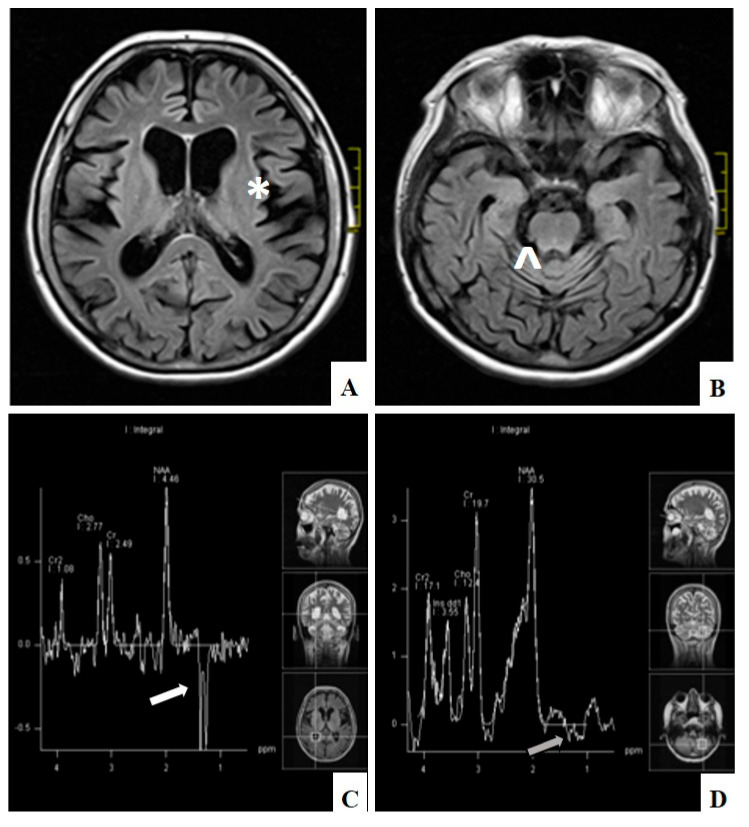
Diffuse cerebral (*) (A) and cerebellar atrophy (^) (B) on MRI, FLAIR and lactate peak) in the lateral ventricle (white arrow) (C) and normal level in the cerebellum (grey arrow) (D) on MRI spectroscopy in patient with ME (patient number 60).

**Figure 4 genes-12-00054-f004:**
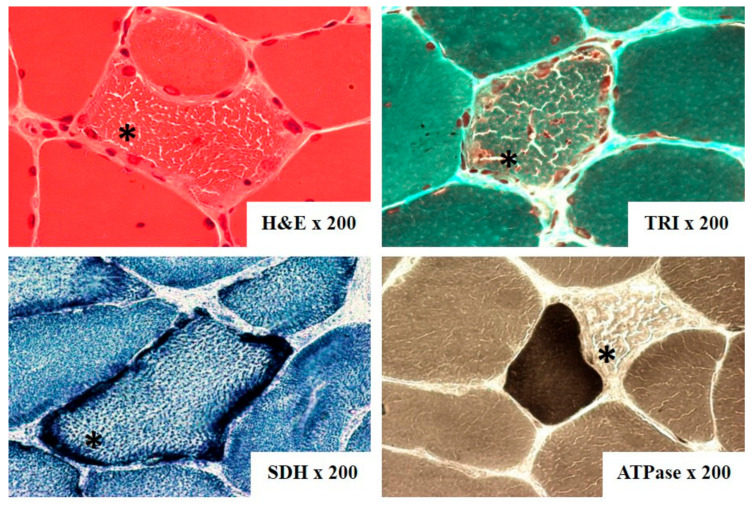
Ragged-red fibers (*) on skeletal muscle biopsy from patient with CPEO+ (patient number 45), (H&E—hematoxylin&eosin, TRI—Gomori trichrome, ATPase—adenosine triphosphatase pH 9.4, SDH—succinic dehydrogenase).

**Figure 5 genes-12-00054-f005:**
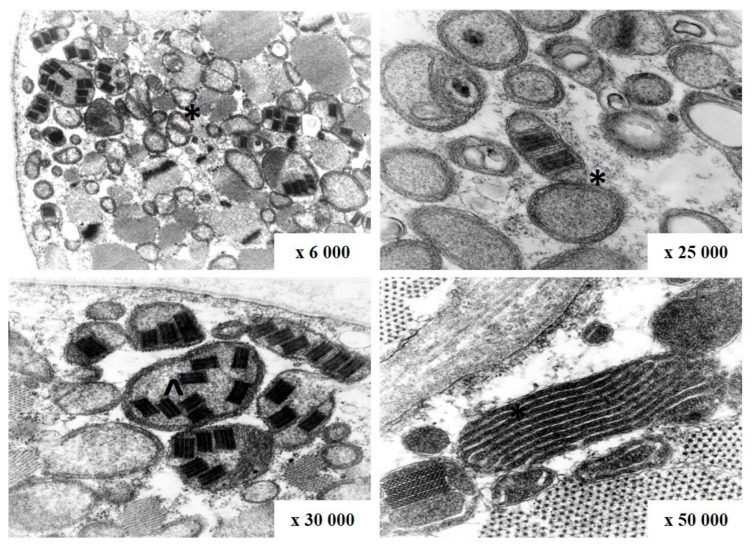
Abnormal mitochondria with irregular cristae (*) and paracrystalline inclusions (^) on electron microscopy in patient with CPEO (patient number 25).

**Figure 6 genes-12-00054-f006:**
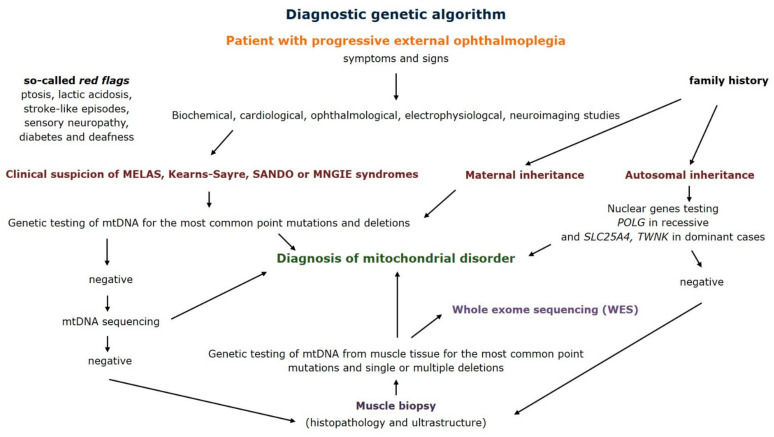
Diagnostic genetic algorithm for patients with progressive external ophthalmoplegia (MELAS—mitochondrial encephalomyopathy, lactic acidosis and stroke-like episodes syndrome, SANDO—sensory ataxic neuropathy, dysarthria, ophthalmoparesis syndrome, MNGIE—mitochondrial neurogastrointestinal encephalomyopathy syndrome).

**Table 1 genes-12-00054-t001:** Characteristics of patients with progressive external ophthalmoplegia.

	CPEO	CPEO+	ME	KSS	SANDO
Number of patients	27	24	26	6	1
Males (M)/females (F)	10/17	8/16	10/16	2/4	-/1
Age M/F	31–78/12–71	33–61/30–76	11–61/18–66	14/15–36	-/51
Mean age of onset	31	35	24	11	31
Mean age of diagnosis	42	50	43	21	51
Positive family history	4	2	4	-	1
CK ranges (U/l)	17–333	16–673	9–1096	13–41	115
Total NCS	24	22	26	4	1
NCS—n	19	18	19	4	-
NCS—A-m	1	-	1	-	-
NCS—A-s	2	-	4	-	-
NCS—A-m + s	2	2	1	-	1
NCS—A-D-m + s	-	1	1	-	-
NCS—D-m + s	-	1	-	-	-
Total EMG studies	23	22	26	4	ND
EMG—n	11	6	9	3	ND
EMG—M	8	13	9	1	ND
EMG—N	3	3	6	-	ND
EMG—M+N	1	-	2	-	ND
RRF/number of biopsies	18/23	17/21	20/24	4/5	1/1
COX-negative/number of stainings	2/6	7/7	8/10	ND	ND

Abbreviations: CPEO—chronic progressive external ophthalmoplegia, CPEO+—chronic progressive external ophthalmoplegia plus, ME—mitochondrial encephalomyopathy, KSS — Kearns–Sayre syndrome, SANDO—sensory ataxic neuropathy, dysarthria, ophthalmoparesis syndrome, M—males, F—females, CK—creatine kinase, NCS—nerve conduction studies, EMG—electromyography, n—normal, A-m—axonal motor neuropathy, A-s—axonal sensory neuropathy, A-m + s—axonal motor and sensory neuropathy, A-D-m + s—axonal and demyelinating motor and sensory neuropathy, D-m + s—demyelinating motor and sensory neuropathy, M—myogenic, N—neurogenic, M + N—myogenic and neurogenic, RRF—ragged-red fibers, COX—cytochrome c oxidase, ND—not done, “-”—no data.

**Table 2 genes-12-00054-t002:** Genetic results in patients with progressive external ophthalmoplegia.

	CPEO	CPEO+	ME	KSS	SANDO
Number of patients	27	24	26	6	1
m.3243A>G	-	-	2	-	-
4977 bp deletion	8	5	3	3	-
Single mtDNA deletion *	6	4	4	1	-
2 mtDNA deletions	2	2	3	2	-
multiple mtDNA deletions	10	11	14	-	1
*POLG* pathogenic variants	2	2	2	-	1
	2 ^multiple^	2 ^multiple^	2 ^multiple^	-	1 ^multiple^
*TWNK* pathogenic variants	2	2	1	-	-
	1 ^multiple^ 1 ^nd^	2 ^nd^	1 ^multiple^	-	-
*RNASEH1* pathogenic variants	-	-	2	-	-
	-	-	1 ^multiple^ 1 ^2^ ^del^	-	-

Abbreviations: CPEO—chronic progressive external ophthalmoplegia, CPEO+—chronic progressive external ophthalmoplegia plus, ME—mitochondrial encephalomyopathy, KSS—Kearns–Sayre syndrome, SANDO—sensory ataxic neuropathy, dysarthria, ophthalmoparesis syndrome, * single mtDNA deletion other than 4977 bp, “-”—no data. In superscript: multiple—multiple mtDNA deletions, 2 del—2 mtDNA deletions, nd—not done.

**Table 3 genes-12-00054-t003:** Number of investigated tissues and results of screening for deletions.

Tissue	Blood	Muscle	Other (Urinary Epithelium, Hair)
all screened	77	71	7
Single, common	12	17	1
Single, non-common	5	11	2
2 del	-	9	0
multiple	1	34	3
negative	59	-	1

Abbreviations: del—deletion, “-”—no data.

## Data Availability

The data presented in this study are available on request from the corresponding author. The data are not publicly available because of ethical and privacy issues.

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
