# Peer review of "Progressive External Ophthalmoplegia in Polish Patients—From Clinical Evaluation to Genetic Confirmation"

_genes, 2020, doi:10.3390/genes12010054_

Round 1

Reviewer 1 Report

In this nice descriptive study, the authors provide the results of mostly targeted genetic analysis in 84 patients with progressive external ophthalmoplegia (PEO) who presented to one clinical centre in Poland over 20 years. They do not describe novel genes or insight, but the results are pretty comprehensive and provide a good overview of the genetic diversity there exists in this typical feature of patients with mitochondrial disease. 

Major comments: 

  • It is not clear to me how CPEO+ is distinguished from CPEO with ME, KSS or SANDO. If this distinction is indeed made clinically, this should be described properly in the methods sections. 
  • It is not clear how "2 mtDNA deletions" (eg in Table 2) is distinguished from multiple deletions and why this distinction is made. If so, this should be explained in the methods section, and it might be worth showing some gel images from this.
  • The discussion is too long, and discussing all clinical syndromes separately is not entirely helpful. One aspect that is missing to me, is a discussion about how the findings in this cohort (clinical symptoms, genetic mutations,...) relate to other research cohorts or the wider literature. Is there a reason to think the population in Poland has a different presentation, or is this mostly a (also valuable) confirmation of what has been observed previously? 

Minor comments: 

  • The abstract is somewhat confusingly structured, but otherwise contains the relevant information. 
  • Introduction: 
    • line 49: one to fifteen copies of mtDNA: reference required
    • line 53: the other non-coding genes should be mentioned, in particular the tRNAs, which are highly relevant for CPEO. 
    • from line 60 onwards: the list of genes is not exhaustive, but is presented as if it were exhaustive. It is not clear which genes are disease genes, and some disease genes are missing. 
    • line 79: apart from red flags, it might be worth referring to the mitochondrial disease criteria (eg Morava et al, Neurology, 2006). 
    • line 86: family history is worth mentioning as important part of the diagnostic work-up. 
  • Methods: 
    • line 102: why were patients randomly chosen for EM? 
    • The PCR parameters might be more appropriate in the supplementary material, together with the primers? 
  • Results: in general, many of the results described in the text would be easier to understand if they are interpreted, together with a reference to the tables for the exact numbers. But this is a matter of preference probably. 
    • line 170: family history could be added to the table. 
    • line 175: CK can be added to the table
    • line 189-199: "few", "most" and "several" patients is too vague, numbers should be provided. 
    • line 190: which cardiological assessment was done?
    • Table 1: for NCS and EMG, total number of studies needs to be specified. 
    • Figure 4: I have no experience with EM studies and cannot review these findings
  • Discussion: see above (major comments)
    • line 315: it is not clear why biceps brachii biopsy is recommended. This should at least be reference or explained. 
    • line 323: it would worth briefly commenting on the current push for first-line whole-exome or genome sequencing, in particular from UK cohorts. It would be interesting to see how the authors see this evolve in their local settings. 
  • References: the numbers do not always match the numbers in the text. Eg line 81: Haas et al is 4, but number 3 in the reference list. 
  • The supplementary information is appropriate and useful. 

Reviewer 2 Report

Kierdaszuk et al describe detailed clinical and genetic characteristics of Polish patients with PEO.

P3 147: I'm not sure what the part that says “Sanger sequencing whole exome sequencing” means, should there be a period after Sanger sequencing?

P4 171-173, Table1: Patients with a family history are included. Are any of the 84 patients family related? If so, the number of families should also be listed in Table 1.

Figure 2-4: These figures are illustrative of each diagnostic method, but it should be indicated which patient the data comes from. Also, areas of interest should be indicated with arrow.

Figure 3: It's supposed to have a magnification of the microscope after the x, but you may miss it.

P8 231-238, Table 2: There is no description of detailed deletion sites, but it should be described in the text or Table S2.

P8 231-238, Table 3: The number of blood samples is 77, but the total number of the breakdown listed in Table 3 is 78. Is this a wrong number or is there an overlap?

P12 323-324: You have done whole exome analysis, but was there any benefit to doing WES in this study? It is not clear whether there were any variants that could not be identified by searching for mtDNA or specific nuclear genes, so this should be accurately described.

Discussion: It is titled “Polish patients”, but there is no discussion of ethnic or regional perspectives, is there anything you can discuss about that?

Table S2: "nd" and "-" are not explained, which is very confusing and can be misleading. For m.3243A>G, the heteroplasmy rate should be described. There is no variant description for TK2, POLG2 and SLC25A4, but this should also be shown. It is helpful to describe which variants are registered as pathogenic in ClinVar, etc.
